# Testing a Model of Pacific Oysters' (*Crassostrea gigas*) Growth in the Adriatic Sea: Implications for Aquaculture Spatial Planning

Camilla Bertolini [1,*], Daniele Brigolin [2], Erika M. D. Porporato [1,3], Jasmine Hattab [4], Roberto Pastres [1] and Pietro Giorgio Tiscar [4]

1   Dipartimento di Scienze Ambientali, Informatica e Statistica, Università Ca' Foscari di Venezia, 30172 Venezia, Italy; erika.porporato@gmail.com (E.M.D.P.); pastres@unive.it (R.P.)
2   Dipartimento di Culture del Progetto, Università IUAV di Venezia, 30135 Venezia, Italy; dbrigolin@iuav.it
3   IMC—International Marine Centre, Loc. Sa Mardini, 09170 Oristano, Italy
4   Facoltà di Medicina Veterinaria, Università degli Studi di Teramo, 64100 Teramo, Italy; jhattab@unite.it (J.H.); pgtiscar@unite.it (P.G.T.)
*   Correspondence: camilla.bertolini@unive.it

**Abstract:** Assessing the potential biomass yield is a key step in aquaculture site selection. This is challenging, especially for shellfish, as the growth rate depends on both trophic status and water temperature. Individual ecophysiological models can be used for mapping potential shellfish growth in coastal areas, using as input spatial time series of remotely sensed SST and chlorophyll-a. This approach was taken here to estimate the potential for developing oyster (*Crassostrea gigas*) farming in the western Adriatic Sea. Industry relevant indicators (i.e., shell length, total individual weight) and days required to reach marketable size were mapped using a dynamic energy budget model, finetuned on the basis of site-specific morphometric data collected monthly for a year. Spatially scaled-up results showed that the faster and more uniform growth in the northern Adriatic coastal area, compared with the southern one, where chlorophyll-a levels are lower and summer temperatures exceed the critical temperature limit for longer periods. These results could be used in planning the identification of allocated zones for aquaculture, (AZA), taking into account also the potential for farming or co-farming *C. gigas*. In perspective, the methodology could be used for getting insights on changes to the potential productivity indicators due to climatic changes.

**Keywords:** Adriatic Sea; AZA; dynamic energy budget; modeling; thermal limits; primary productivity; satellite data

## 1. Introduction

Aquaculture could play a crucial role in meeting rising food demand [1–3]. To this end, bivalve shellfish culture is a sustainable option [4], as most commercial shellfish species are fast growing, can be farmed at relatively high densities and have no needs for additional feed [5,6]. Moreover, shellfish farms can contribute to maintaining essential ecosystem functions [7] such as nutrient cycling [8], fitting both into UN SDG 2, the blue carbon initiative [9], and the EU blue growth strategy (COM494/2012). In the Mediterranean region, the identification of allocated zones for aquaculture (AZA) is used to guide and sustain the development of the industry (GFCM/36/2012/1), and the science-based identification of proper areas for aquaculture within maritime spatial planning is perceived as a priority among stakeholders [10].

*Crassostrea (Magallana) gigas* is considered a cosmopolitan species, able to withstand a wide temperature range [11] This species has been recorded in Japan, Korea, Siberia, Australia, the United States and Canada. In North America, its geographical range spans from Southeast Alaska to Baja California, and in Europe from the British Isles to Portugal and

the Mediterranean. While in Europe it is an invasive species, its cultivation is widespread, and it can be now considered as "naturalized" in many areas [12]. France is, by far, the largest European producer with 115,000 tons annually (FAO 2019). In the Mediterranean, cultivation of this species accounts only for 3% of the volume of shellfish production [13], and the production of *Ostrea edulis* is negligible. An increase in oyster production would benefit the local economy, as their wholesale price (3.5–6 €/kg) is around 5.5 time that of mussels [14]. The Adriatic Sea is, at present, the most important Mediterranean area in terms of mussel production in longline systems (22 metric tons in 2013, ~33.6% of the Italian production, MiPAAF 2014), and represents an area for potentially developing oysters' cultivation up to 6 km offshore [14]. However, the current oyster production is still limited to small-scale farming, with ca. 53 tons produced in 2013 (FAO 2019).

Most modeling and field studies on oyster growth and survival have been focusing on Atlantic sites [15,16], which present water temperature and trophic levels markedly different from those in the Adriatic Sea. In particular, chlorophyll-a levels are significantly lower in the Adriatic Sea compared to other Atlantic coastal zones where this species is typically cultured [17]. Moreover, the water temperature rarely exceeds 21 °C in Atlantic waters, which is around *C. gigas'* optimum, while summer temperature in the Adriatic Sea can reach 30 °C, near the identified critical upper thermal limit for this species of 32 °C [18,19]. Assessing the growth performance of this species in a region close to its thermal limit and with a lower primary productivity is therefore important to understand whether expanding oyster culture can be a viable option.

Individual-based growth models can be used as tools to estimate a set of indicators, such as time to harvest and average size at harvest, in relation to site-specific time series of forcing functions: for shellfish the most important ones are water temperature and chlorophyll-a, which is taken as a proxy of energy available to filter feeders. The above forcings can be estimated over large coastal areas from remotely sensed data; therefore, potential productivity indicators can be mapped and used in aquaculture site selection [15,20,21]. Dynamic energy budget (DEB) theory [22] has provided a framework for developing robust individual ecophysiological models [23], based on a limited set of assumptions and applicable to a wide range of species, including farmed ones (e.g., [24]). The large amount of satellite data currently available allows one to run ecophysiological models in a spatially explicit manner [15,21].

DEB models were previously applied to investigate and predict *C. gigas'* growth in both North East Atlantic [25,26] and Mediterranean regions [27,28] but, to our knowledge, they were not previously applied to the Adriatic. Even though the parameterization of DEB is robust and soundly based on well-established theoretical framework, their application to new areas usually requires the finetuning of a parameter, namely the half-saturation coefficient, which depends both on the species and its diet. For filter feeders, this parameter represents the concentration of feeding particles at which half of the maximum intake rate is reached [29,30]. In order to assess the potential productivity of *C. gigas* in the western Adriatic coastal areas, a comprehensive set of field data was collected and used for finetuning an individual model. The model was subsequently applied in mapping a set of indicators, using as input a decadal spatial time series of environmental forcings.

## 2. Methods

### 2.1. Model Parameter Estimation

The individual growth of *C. gigas* was simulated using the model presented in [31], which, besides the critical upper thermal tolerance limit for respiration of 32 °C [26], includes a critical ingestion upper limit (25 °C). This feature was considered more consistent with ecophysiological studies, which showed a decrease in feeding capacity at temperatures above 25 °C, which is lower than the metabolic temperature limitation typically set at 32 °C [32]. DEB models include a set of parameters, which may require site-specific tuning, as the same species can show adaptation to different environmental conditions [33]. The estimation of site-specific parameters is, therefore, very relevant for subsequent application

of the model to areas in which it was not previously tested. Based on the data presented in Section 2.2, the half-saturation coefficient ($X_k$), which depends on food quality, was estimated [30].

The estimation of the $X_k$ parameter was performed with a bootstrap methodology [34], and using the mean squared error (MSE) as a goal function to compare estimates of oyster sizes obtained with different values of the parameters with empirical data:

$$MSE = \sum_{j=1}^{n} \left( \frac{\left( \frac{W_m - \hat{W}_m}{\sigma_W} \right)^2 + \left( \frac{L - \hat{L}}{\sigma_L} \right)^2}{n} \right) \quad (1)$$

where $n$ is the number of sampling events, and $W_m$, and $L$ represent, respectively, the values of meat wet weight and the shell length randomly extracted from a synthetic population of 2000 individuals, $\sigma_W$, $\sigma_L$, are the standard deviations of the observed data, while $\hat{W}_m$ and $\hat{L}$ are the respective model predictions. The minimization of MSE was independently performed on the 2000 synthetic growth curves, which were generated by randomly extracting $W_m$, and $L$ values from a probability density function which was assumed to be normally distributed, with mean and standard deviation defined by the descriptive statistics of the in situ observations at the $n$-different sampling events.

The shape coefficient used in the model was in this study calculated from the data collected in situ, as $V^{1/3}/L$. V was calculated assuming the oyster geometry to be made of two ellipsoid-based cones, one taller (3/4 height) and one flatter (1/4 height). The mean value and standard deviation were calculated, and the mean was then taken for use in the model (see Table 1). Total weight was calculated from wet meat weight through an allometric relationship, with the coefficients estimated by performing a linear regression based on empirical oyster data (see Table 1).

**Table 1.** Dynamic energy budget (DEB) parameters used in this study. In bold are the parameters estimated from the experimental data. *: Cultivated oysters are triploid. In this study sterility is assumed.

| Parameter | Value | Reference |
| --- | --- | --- |
| Arrhenius temperature $T_A$ | 5800 K | [35] |
| Rate of decrease at lower boundary ($T_{AL}$) | 75,000 K | [35] |
| Rate of decrease at upper boundary ($T_{AH}$) | 30,000 K | [35] |
| Critical lower limit ($T_L$) | 276 K | [31] |
| Critical ingestion upper limit ($T_H$) | 298 K | [31] |
| Critical respiration upper limit ($T_{H1}$) | 305 K | [31] |
| **Half-saturation coefficient ($X_K$)** | **12.6** | **This study** |
| Max. surface area-specific ingestion (JXm) | 560 J/cm$^2$ d | [35] |
| Assimilation efficiency (ae) | 0.75 | [35] |
| Volume-specific maintenance costs (p_M) | 24 J cm$^3$/d | [35] |
| Max. reserve density ($E_m$) | 2295 J/cm$^3$ | [35] |
| Cost for growth ($E_g$) | 1900 J/cm$^3$ | [35] |
| Energy content of reserve (mu E) | 17,500 J/cm$^2$ | [36] |
| Allocation fraction to somatic tissue (k) | 1 | * |
| Volume at puberty ($V_p$) | n/a | [35] |
| Reproduction efficiency ($K_R$) | 0 | * |
| Gonadosomatic index (GSI) | 0 | * |
| **Shape coefficient ($\delta$m)** | **0.225 ± 0.03** | **This study** |
| Wet meat weight to dry meat weight converter | 0.2 | [37] |
| **Dry meat weight to total weight equation** | **23.8 W$_{md}$ + 6** | **This study** |

## 2.2. Mapping Indicators

Three growth-performance indicators used by the producers, namely shell length (L), total weight (W) and the number of days required to reach the marketable size of 6 cm (TTM—time to market), were selected [38,39].

In order to obtain robust estimates of the above indicators, which take into account the interannual variability in temperature and primary production, the model was run for nine 28 month-long "grow-out" cycles, (see Section 2.3), spanning from 2008 to 2019. The simulated domain covered a distance from the coastline corresponding to the 40 m isobath, extending from Friuli Venezia Giulia up to Puglia region, with a 1 km$^2$ resolution. The DEB model was forced at each cell of the grid, using time series of SST and chl-a data estimated from satellite data, as described in Section 2.4. Marketable size was not always achieved, thus for the TTM indicator, the median and interquartile ranges were calculated by considering only those years in which the final length of 6 cm was achieved. Cells of the grid in which the marketable size was achieved in all the 9 grow-out periods were named "persistently suitable growing area", and marked on the maps.

All model runs were performed in MATLAB 2020b [40]. Statistical analyses on the sampled data were performed using R [41] and maps were generated using QGIS 3.12.2 [42].

## 2.3. In Situ Data Collection

In order to estimate the parameter $X_k$, time series of shell length, width, height (thickness), total wet weight and meat wet weight were collected in situ during part of a grow-out cycle of triploid oysters, *Crassostrea gigas*, in an oyster farm located in the southern Adriatic Sea, located near the Capoiale estuary, Cagnano Varano, Foggia, Italy (41°56.469′ N, 15°41.539′ E). Farm specifications and data collection are described below. The grow-out cycle started in December with individuals of a T4 size (0.6 cm) and ended after 28 months, following procedures from the sampled farm (see below). This was similar to other studies and areas (e.g., [37,41,42]).

The dataset was collected at a farming site managed by the cooperative Varano La Fenice (Cagnano Varano, Foggia, Italy). This company employs SEAPA baskets (dimensions: 650 × 400 mm$^2$; volume: 24 L) in polyethylene copolymer (model MP650) in piles of four baskets each, held at a depth of 6 m. *C. gigas* seed (size T4–T6) imported from France, was stocked in December 2016. Stocking density at the beginning of the sampling was 201 oysters/basket. The same number of individuals (60 specimens) was collected every month from each of the four baskets, from March 2018 to February 2019. Eleven surveys were carried out, as weather conditions were adverse in November 2018. The initial sample size, 60 specimens, was reduced to 40 from August 2018, to limit the effects of sampling on stocking density (Table A1 in Appendix A). The following biometric parameters were determined on a subset of 15 individuals: shell length, width, height (thickness), total wet weight and meat wet weight. Mortalities were recorded when oysters appeared empty or remained permanently open.

## 2.4. Forcing Functions

The DEB model requires as inputs daily time series of (i) water temperature, (ii) and chlorophyll-a (chl-a) concentration, as a proxy for living phytoplanktonic cells that can be cleared and digested by shellfish. In this study, these forcing variables were estimated from time series of satellite remote sensing (SRS) daily data at 1 km$^2$ spatial resolution, obtained from the Copernicus Marine Environment Monitoring Service (CMEMS; https://marine.copernicus.eu/ accessed on 15 January 2020) EU program. Daily sea surface temperature (SST) Level 4 (i.e., continuous spatiotemporal data resulting from model outputs) and chl-a Level 3 (i.e., data mapped on uniform spatiotemporal grid but with gaps due to clouds presence), were used. The chl-a dataset was linearly interpolated, in order to fill unequally spaced parts of the time series, thus obtaining two matrices of daily chl-a and SST data of equal size. For the model calibration, the chl-a and SST relative to the point coordinates of the farm were taken for the grow-out cycle related to the sampling

(December 2016–March 2019). Subsequently, the median and interquartile range (IQR) of both SST and chl-a were mapped in order to evaluate their variability within the 9 grow-out cycles (28 months from December to March, in the 11-year time series spanning from 2008 and 2019). Moreover, considering the SST forcing, we calculated the number of days in which the temperature exceeded the critical upper ingestion limit ($T_H > 298$ K).

## 3. Results

### 3.1. Parameter Estimation

Sea surface temperature and chl-a time series at the farm site, for the grow-out time period 2016–2019, corresponding to the time of the in situ data collection, are shown in Figure 1. SST ranged from 9.5 to 29.1 °C and chl-a from 0.14 µg/L to 7.85 µg/m with peaks in winter and spring months (December to March), and lowest values during the summer months. During the grow-out cycle, SST outside the respiration tolerance range of oysters, i.e., above 32 °C and below 3 °C, were not observed. However, SST exceeded the ingestion rate upper tolerance limit of 25 °C for 71 days in 2017 and 98 days throughout the summer of 2018 (Figure 1).

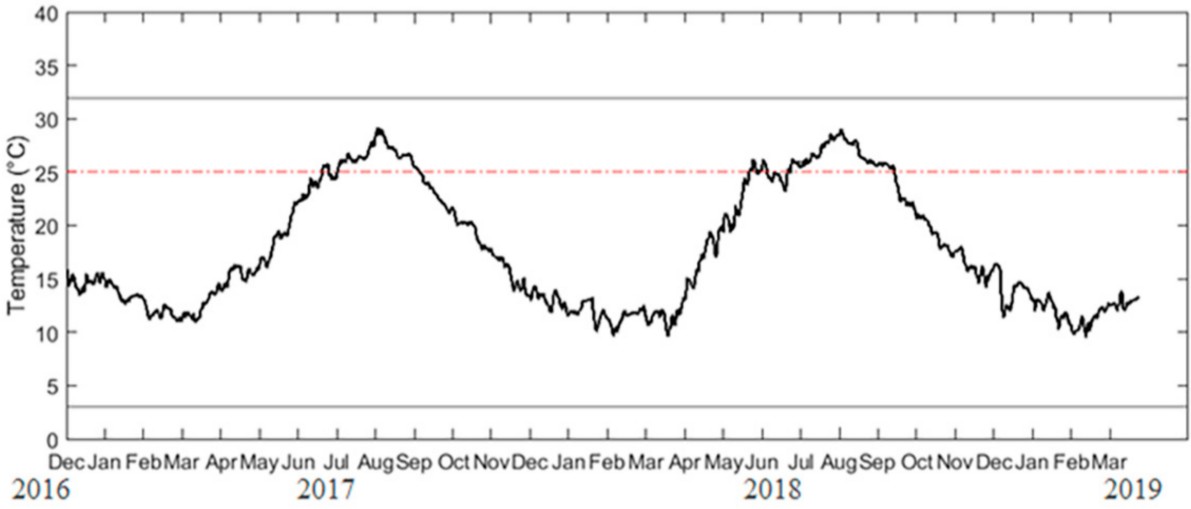

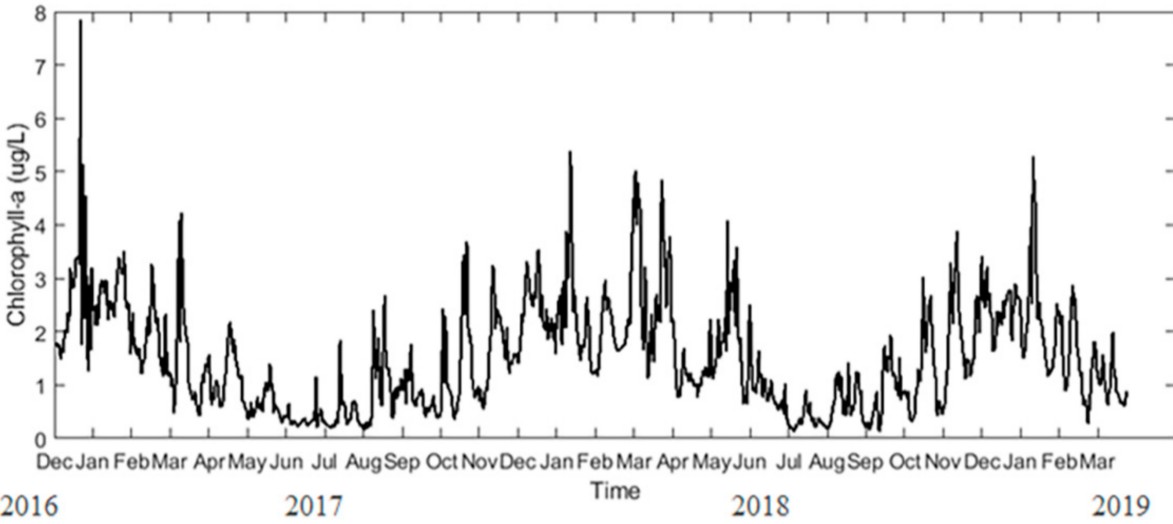

**Figure 1.** Temperature (°C) and chl-a µg/L for the 2016–2019 grow-out period, at the study site, used as forcing functions in the DEB model. Horizontal solid lines represent lower and upper critical thermal limits. Horizontal dotted red line represents thermal limits for feeding.

All model parameters are summarized in Table 1. The calibration of the half-saturation coefficient, $X_k$, gave a value of $12.6 \pm 1.3$ (µg chl-a $L^{-1}$, median $\pm$ iqr, Figure 2).

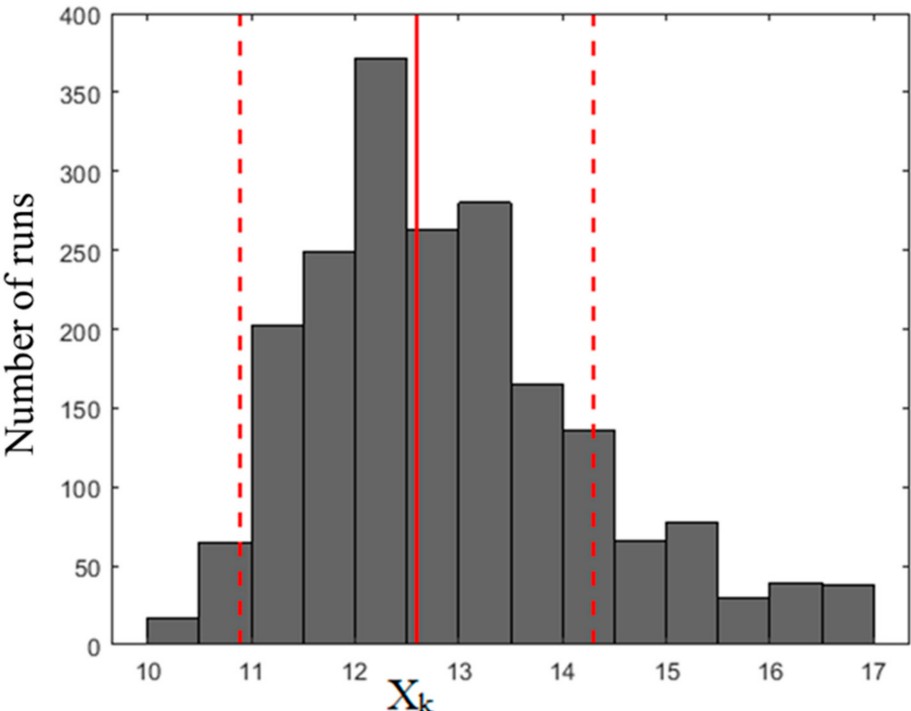

**Figure 2.** Histograms of $X_k$ resulting from the bootstrap simulation. Solid red line represents median, dashed line represent interquartile ranges of the value.

The model output for the whole grow-out period using the median value of $X_k$ (estimated within this study) is compared with the observations at the Cagnano–Varano farm in Figure 3, which shows the comparison between simulated and observed length and meat wet weight (Figure 3a,b) and the estimated and observed total weight (Figure 3c). Overall, the model fits the growth patterns, providing accurate predictions (model line falling well within the observation standard deviations) of all variables from the start of sampling in March (15 months from seeding) up to December (24 months from seeding) but underestimates the growth in terms of weights from January (month 25) to March (month 27) 2019 (Figure 3b,c).

### 3.2. Spatial Distribution of Indicators

The environmental variables used to run the DEB models for the whole Italian Adriatic basin are mapped in Figure 4. SST presented the highest median values in the southern part of our study area, with a marked latitudinal gradient (Figure 4A), and the highest temporal variability within the 11 years considered, in the northern part, in the Veneto and Emilia Romagna regions (Figure 4B). Chl-a had the highest median values and variability recorded in the area surrounding the Po river outlet ($3 \pm 2.6$ mg $m^{-3}$), with values decreasing moving from this area and the lowest values recorded on the coasts around the Apulian region ($0.9 \pm 0.7$ mg $m^{-3}$) (Figure 4C,D). Chl-a values decreased moving offshore (down to $0.2 \pm 0.1$ mg $m^{-3}$).

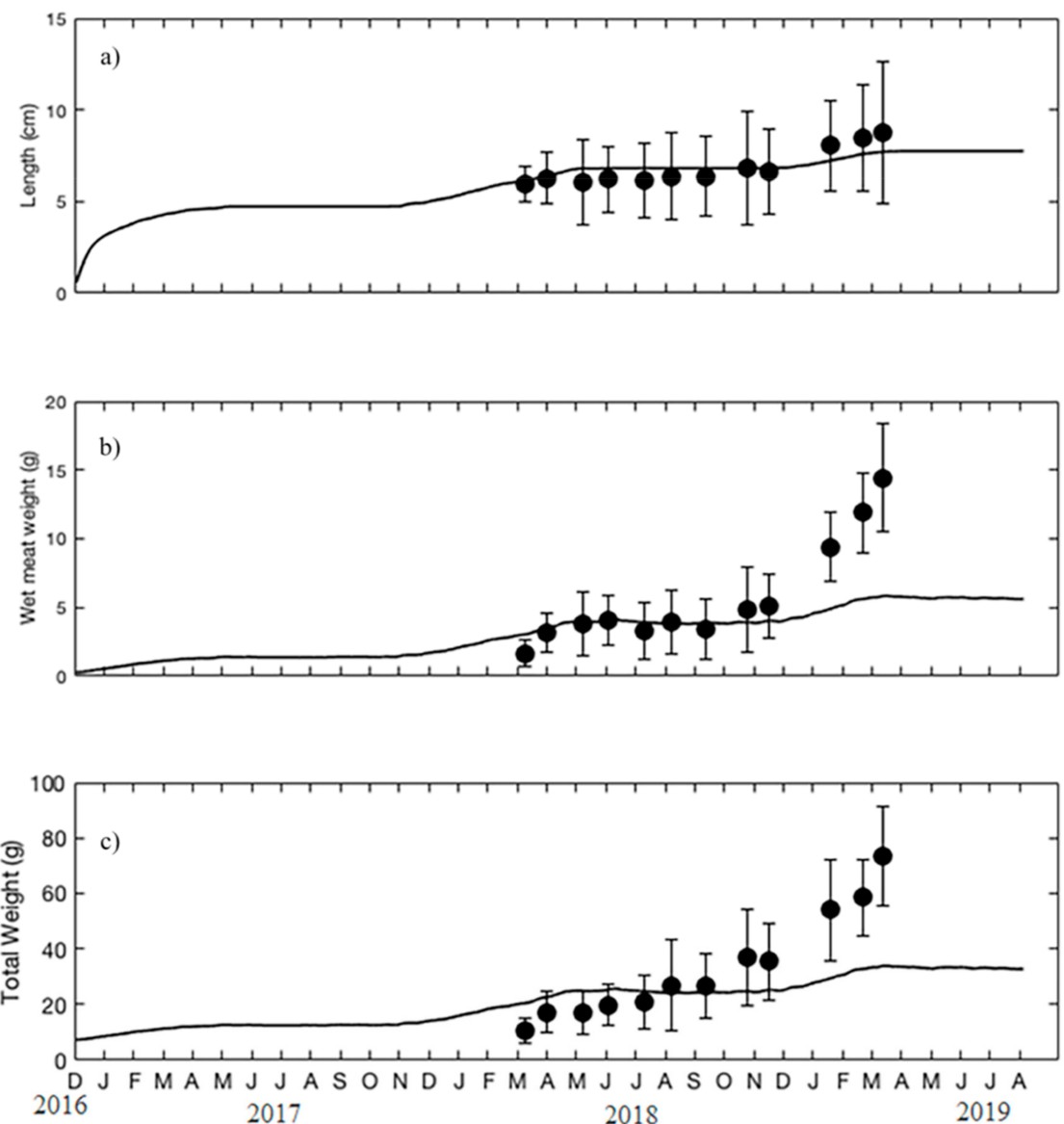

**Figure 3.** Output of DEB model and observations, (**a**) shell length (cm), (**b**) wet soft-tissue weight (g), and (**c**) the estimated total weight (g) from beginning December 2016 to end of July 2019. Dots represent sample means and error bars represent sample standard deviations.

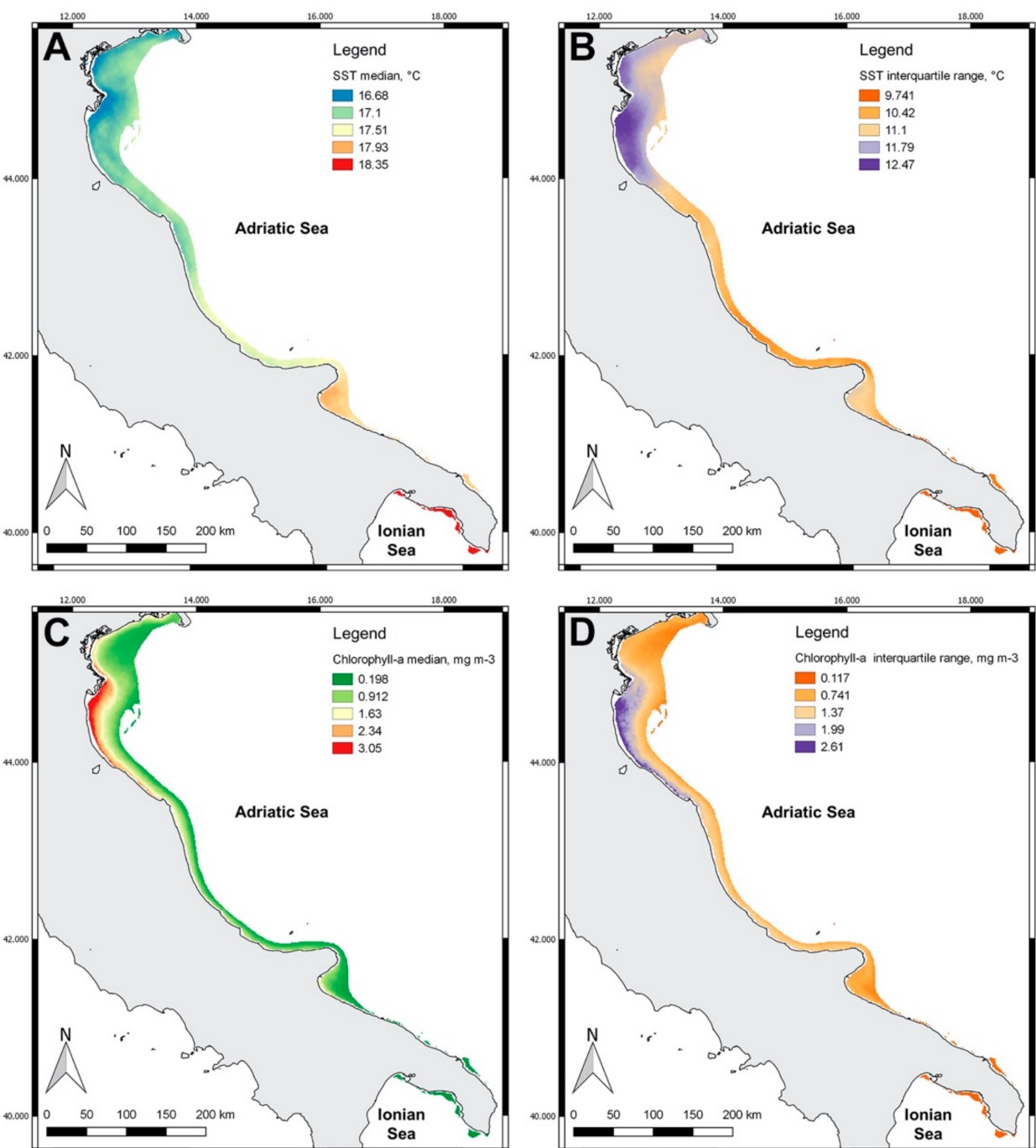

**Figure 4.** Environmental variables: (**A**) median, (**B**) sea surface temperature (SST), interquartile range, (**C**) chl-a median, (**D**) chl-a interquartile range.

Figure 5 shows the number of days in which temperatures exceed the upper ingestion limit set by Bourlés et al. (2009), which shows a latitudinal gradient with increased number of days moving south (from less than 670 days up to 978 in the 2008–2019 period).

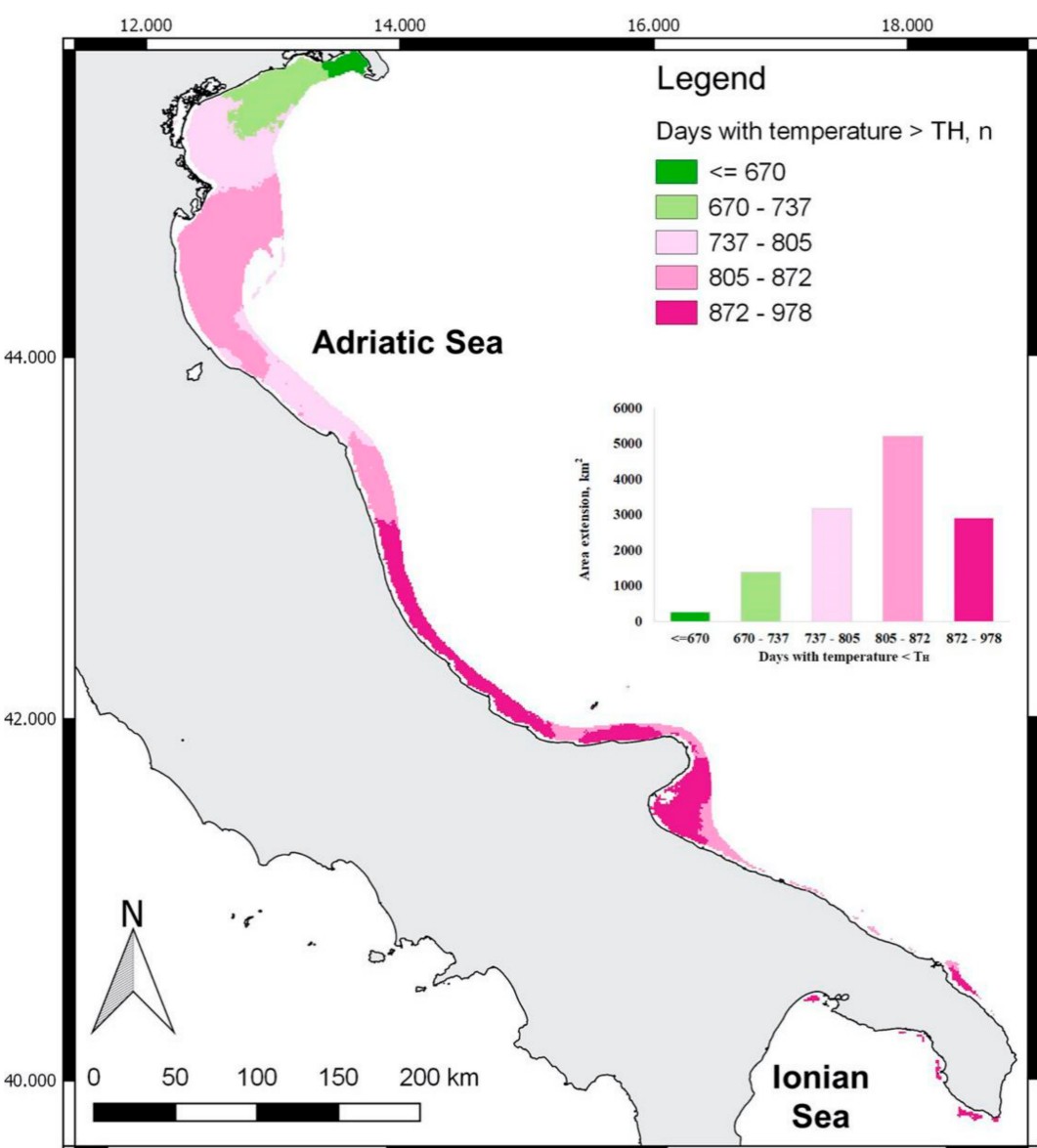

**Figure 5.** Number of days during the entire period considered (2008–2019) with temperature > $T_H$. and area extension (km$^2$) for each day's class.

Figure 6 shows the spatial distributions of the median and the IQR of the growth indicators: L, W and TTM. The median length at the end of each grow-out cycle ranged from 2.75 ($\pm$0.04) to 18.96 ($\pm$5.18) cm. Oysters were longer in the areas affected by the Po River discharge, with a tendency to be shorter moving offshore (down to 3 cm). This area was also characterized by the largest IQR variability between grow-out periods (up to 2.9 cm) while the lowest length values and lowest variability were predicted in the Apulian region (6 cm near the coast) (Figure 6A,B).

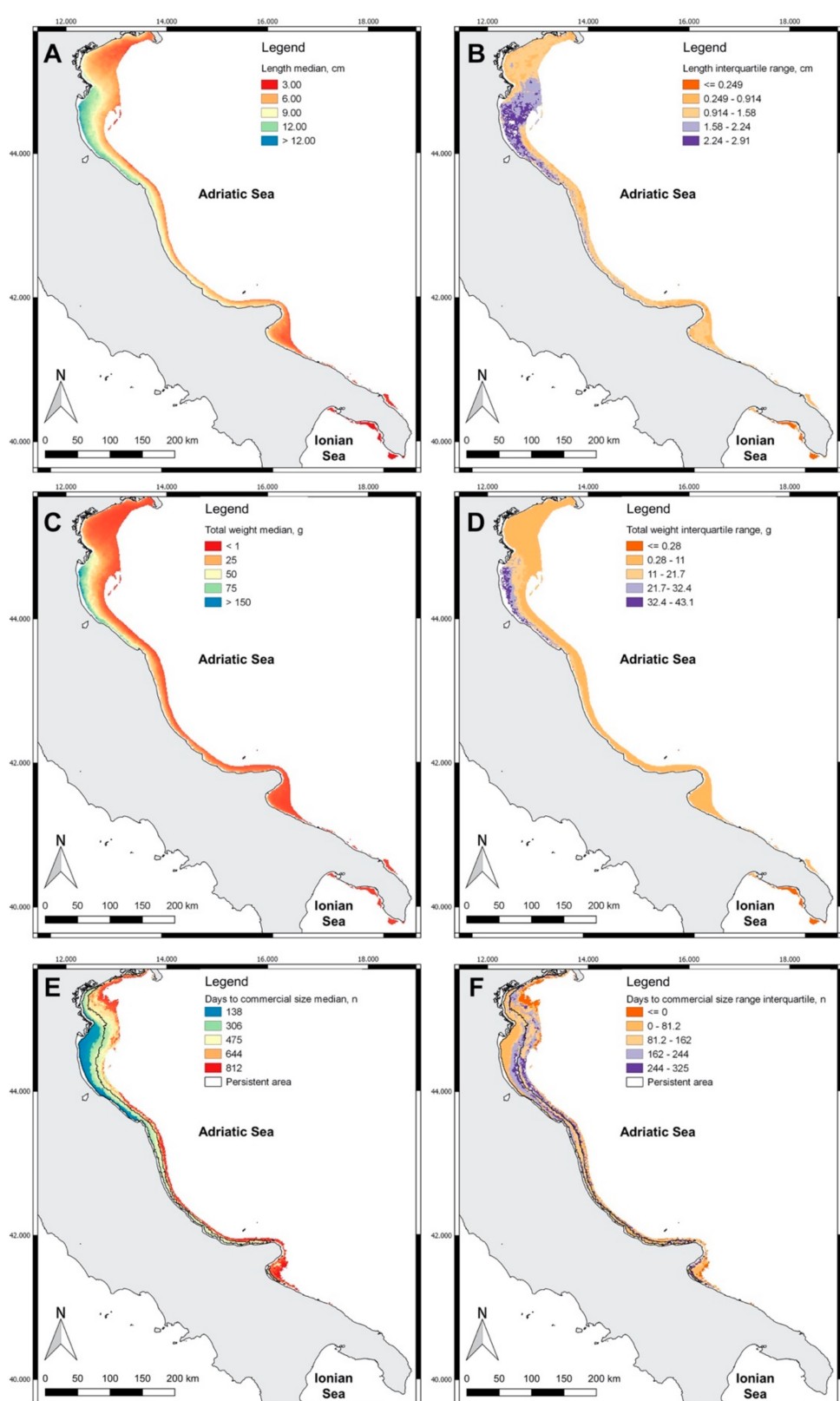

**Figure 6.** Oyster farming indicators: median length (**A**) and interquartile range (**B**); total weight median (**C**) and interquartile range (**D**); days to reach the marketable size median (**E**) and interquartile range (**F**).

The median total weight reached at the end of the simulated grow-out cycle, and its variability, are mapped in Figure 6C,D. This value showed a pattern similar to the length

one, with the highest median values and variability (up to >150 ± 43 g) recorded near the Po River outlet, and a decreasing tendency moving offshore (down to <1 g).

TTM median and interquartile ranges (Figure 6E,F) present the lowest values (138 days) in the northern inshore areas, while in the southern and offshore parts it took a greater number of days (475 southern inshore and 812 offshore) to complete the grow-out cycle. TTM predicted by the model in the area where the sampling was conducted was 422 (median over the nine grow-out periods), which was consistent with 458 days found during the sampling. The TTM largest IQR was found in the offshore area nearby the Po River outlet (325 days). The "persistently suitable growing area", is shown in the maps (Figure 6E,F), and was spread all over the domain considered, from the Venice lagoon down to the Puglia region.

## 4. Discussion

This study aimed to apply a DEB model at a basin scale to inform spatial planning on expanding oyster culture in the Adriatic Sea. This was pursued via the application of an individual-based model, reparametrized for the trophic conditions at the basin scale using remote-sensing data. The results from this study provide some useful advances to the field, including much needed updated information on oyster growth dynamics for a coastal zone such as the Adriatic Sea, where spring and summer water temperatures are increasing at a faster rate compared to those in other areas of the Mediterranean Sea [18,43,44].

The suitability and ubiquity of DEB model applications for *C. gigas* was already demonstrated for multiple Atlantic sites, where SST ranges between 6 and 24 °C [25], i.e., within the thermal tolerance ranges for both ingestion 3–25 °C and for respiration (3–32 °C; Bourlés et al., 2009). While a DEB was previously used at a Mediterranean site located in the Southern Tyrrhenian Sea (e.g., [27]), and also in the Adriatic [28], the model was not previously recalibrated to take into account the site specificity of $X_k$ [29]. The result of the finetuning of $X_k$ indicates that the Southern Adriatic has a food quality slightly lower than that in other Mediterranean areas with similar average chl-a levels (e.g., $X_k$ = 9.5 µg L$^{-1}$ used in [27]). The area with the highest chlorophyll-a levels, in the northern Adriatic, was also the area with the best growth, in which the higher production areas of *C. gigas*, Emilia Romagna and Veneto, are also located [43]. The Adriatic Sea was found to have a highly irregular variance in chl-a concentration compared to other Mediterranean regions in which the phenology of phytoplankton blooms followed more regular patterns [44]. Interannual variability in inorganic seston quantity and quality should also be considered, for example due to differences in riverine nutrient inputs between wet and dry years [45], and within our study area for the nine simulated grow-out cycles, from 2008 to 2019, there was a high spatial variability in chl-a concentrations observed, with median values comprised between 0.13 and 5.34 µg L$^{-1}$, with a variability of 0.07 to 5.14 µg L$^{-1}$. This is in line with the results obtained in a previous study [28] and may represent a key decisional point for allocation of coastal zones to different activities.

It is important to mention that intra-annual variability in food quality, which has not been taken into account using this approach, may occur. This may be one of the factors explaining some of the discrepancies seen in the second winter (January–March 2019) between observed and predicted weights in the model calibration. Food quality might display a seasonal variation that is caused by a succession of plankton species (e.g., [46]). *C. gigas* is able to filter most phytoplankton particles, aside from pico-particles [47], however, studies on the energy obtained by different planktonic functional groups are limited. The seasonality in the lower Adriatic Sea results in the greatest abundance during the winter months (January–February), with a dinoflagellate-dominated community in winter, transitioning through a mixed assemblage of phytoflagellates and diatoms in spring and summer, and returning to a mixture of phytoflagellates and dinoflagellates in autumn [48]. Moreover, in the short term, water density and mixing, which depend on temperature and meteorological conditions (Mediterranean Basin: [49]), can also affect plankton vertical distribution, contributing to its interannual variability. Plankton cell densities [31], or

planktonic carbon [45], would certainly be better proxies of the energy available to shellfish and may result in a better fit in the last section of the grow-out cycle. Models based on DEB theory could be thus potentially improved through the addition of a seasonal term that takes into account the variability in food availability, uptake and energy reserves. Current research efforts in the field of remote sensing are aimed at identifying main planktonic groups, however, to date, chl-a is the only proxy which can be used in site selection studies, as it can be reliably estimated from satellite data [50–52] and, therefore, used for mapping the growth potential at large spatial scales.

The area showing the greatest chl-a concentration and variability is also the area which showed by the highest SST variability (see Figure 2), and this is where the highest variability in growth parameters was recorded. Therefore temperature may also play an important role for the successful farming of this species. Considering that *C. gigas* in the Adriatic Sea lives near its critical thermal limits, even a small increase in water temperature can have direct effects on the growth and survival of this species. Considering the length achieved at the end of the simulated rearing cycle, the areas with the highest variability (Figure 6) almost overlapped with the area characterized by temperature values often exceeding the critical ingestion upper limit ($T_H$) (Figure 5). No growth, or even decrease in wet weight, was observed at the sampling site between July and September 2018, when water temperatures often exceeded the 25 °C threshold identified by [31]. This result is consistent with that presented in [53], who compared two grow-out cycles: the first one characterized by summer temperatures >25 °C, in which individual growth stopped, and a second one with temperatures lower than 22 °C, in which oysters continued growing throughout the summer. Nonetheless, our results are not entirely consistent with previous work on oyster growth in the Adriatic Sea, since [54] observed maximum weight increment between July and October. However, compared to the present study, temperatures were markedly lower (24.5 ± 1 °C vs. 26.2 ± 0.79 °C in July down to 16.4 ± 1 °C vs. 20.4 ± 1.7 °C in October), which may explain the discrepancies. This could be related to the flow of freshwater from the rivers, mainly the Po, located in the northern portion of the basin [55]. Indeed, comparing the flow annual mean values with the number of days with SST > $T_H$ there was an opposite trend, but the extent of the impact of this mechanism needs to be further investigated in a dedicated study.

The model recalibrated in this study was adapted to take account of the triploid nature of the cultivated oyster. Triploidy induction in shellfish aims to obtain faster growth and sterility of reared individuals, and in this study we assumed full sterility as no gametogenesis. However, this is not necessarily always the case, as some individuals may still be producing gametes [56]. Furthermore, triploidy may lead to further biological changes beyond reproduction, for example, differences in the seasonality of immune responses [57] but also differences in net energy balance [58], which may explain some of the discrepancies between the model and the observations in the final period of cultivation, and should be taken into account with ad hoc studies complemented by further biological understanding.

## 5. Conclusions

This study provides a showcase of existing resources supporting spatially explicit approaches for the designation and management of aquaculture areas [59]), combining satellite data and ecophysiological models, supported by in situ sampling. Moreover, the present application could provide a relevant contribution in the broader framework of maritime spatial planning—MSP directive 2014/89/EU implementation [60]. Timely and science-based aquaculture zoning is relevant not only for guaranteeing an effective integration among different sectors considered in MSP (e.g., providing the knowledge base for comparing different scenarios of use), but also from the perspective of land–sea interactions. In fact, shellfish aquaculture depends on the input of energy and materials from the land (e.g., [61]); it can limit local eutrophication [62] but, on the other hand, is sensitive to potential sources of pollution directly related to land originated drivers [63]. Along with local drivers, climate change is expected to have an influence on the environ-

mental conditions of cultivation areas. In this respect, an increased frequency of heatwaves in the Mediterranean Sea [64] could represent a good example of climate change effect, which should be considered carefully when designing areas and addressing risks for oyster cultivation in the Adriatic Sea [65] considering the predicted temperature increase for the this basin [66]. We believe that the increasing availability of models, such as the one applied in the present study, able to combine nonlinearly the effects of water temperatures and trophic conditions, will represent a key resource for predicting future changes in productivity associated with modified water biogeochemical characteristics (e.g., induced by modified thermal conditions coupled with changes in plankton phenology). The variability in growth indicators within the time frame analyzed in this study provides an indication of the robustness of model results, which should be taken into account in the future design of allocated areas, and site selection processes aimed at maximizing resource use, thus ideally prioritizing areas of cultivation where returns are higher and consistent. Indeed, the average growth index among the nine time periods gave only preliminary information about the performance of this organism in the Adriatic Sea, which should be complemented with the collection of further oyster growth data under a range of contrasting environmental conditions experienced by the Adriatic Sea waters.

**Author Contributions:** Conceptualization, C.B., R.P., P.G.T.; methodology, C.B., R.P., E.M.D.P., D.B.; software, C.B., E.M.D.P., D.B.; formal analysis, C.B., E.M.D.P.; investigation, C.B., P.G.T., J.H., E.M.D.P.; resources, P.G.T., R.P.; data curation, C.B., J.H., E.M.D.P., P.G.T.; writing—original draft preparation, C.B., E.M.D.P., D.B.; writing—review and editing, C.B., E.M.D.P., D.B., J.H., P.G.T., R.P.; visualization, C.B., E.M.D.P.; supervision, R.P.; funding acquisition, R.P. All authors have read and agreed to the published version of the manuscript.

**Funding:** The research leading to these results has received funding from the European Union's HORIZON 2020 Framework Programme under GRANT AGREEMENT NO. 773330.

**Institutional Review Board Statement:** Not applicable.

**Informed Consent Statement:** Not applicable.

**Data Availability Statement:** The data presented in this study are available on request from the corresponding author.

**Conflicts of Interest:** The authors declare no conflict of interest.

## Appendix A

**Table A1.** Sampling schedule during the monitored grow-out. Month of sampling, number of oysters present before sampling, number of oysters removed, number of dead oysters, number remaining, average number of oyster in each of the four baskets, approximate volume of water per oyster.

| Month | Number before Sampling | Oysters Taken | Mortalities | Number after Sampling | Number in Each Basket (Average) | Volume of Water per Oyster (L) |
|---|---|---|---|---|---|---|
| March | 864 | 60 | | 804 | 201 | 0.119 |
| April | 804 | 60 | | 744 | 186 | 0.129 |
| May | 744 | 60 | | 684 | 171 | 0.140 |
| June | 684 | 60 | | 624 | 156 | 0.153 |
| July | 624 | 60 | 2 | 562 | 140.5 | 0.170 |
| August | 562 | 40 | 20 | 502 | 125.5 | 0.191 |
| September | 502 | 40 | 67 | 395 | 98.75 | 0.243 |
| October | 395 | 40 | 9 | 346 | 86.5 | 0.277 |
| December | 346 | 40 | | 306 | 76.5 | 0.313 |
| January | 306 | 40 | | 266 | 66.5 | 0.360 |
| February | 266 | 40 | | 226 | 56.5 | 0.424 |

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
