# Peer review of "Testing a Model of Pacific Oysters’ (Crassostrea gigas) Growth in the Adriatic Sea: Implications for Aquaculture Spatial Planning"

_sustainability, doi:10.3390/su13063309_

Round 1

Reviewer 1 Report

The manuscript includes valuable information of oyster culture in vast area applicable for marine spatial planning. Unfortunately, The manuscripts should be added some more information and discussion.

  1. Judging from Fig. 3, difference between predicted and measured (actual) values are different from January through March in weight, I could not find the discussion about this difference. These differences  might start from October, just after summer, spawning season. As the oysters are triploids, are there any possibility that the model is not fully reflect the biology of triploid oyster, even though the allocation of energy for reproduction set to be zero.
  2. The influence from climate change is not discussed in discussion part”, but in conclusion, L333-. As the authors mention about the climate change in the abstract, L24-25, such discussion should be done in the discussion part.
  3. The conclusion part should be included more information and insights for aquaculture spatial planning.

L102-103 The estimation is for Wm and L, not for Xk. The relationship between Xk and (Wm, L) should be presented.

L113  I could not find the way to determine V.

L194   In Table 1, please clarify the definition of * on Triploid oysters. I suggest the authors to add the information that cultured oysters in the area are triploid.

L350-L387  This part should be completed.

Fig. 1 and 3, It is better to add indication of years at X axis.

Fig. 4-6, Legend -> Legend. I think the units, cm, g, etc. should be in the parentheses.

Fig. 5, The title of histogram in the figure should be such as Areal extension of ….

Reviewer 2 Report

Line 72: “Dynamic Energy Budget”: it is recommended to introduce the “DEB” here because, in the following description, the abbreviation DEB is used.

Line 77: It is recommended to use “Pacific oyster” or “Crassostrea gigas” uniformly in the manuscript, as mixed-use may cause confusion.

Line 102: The estimation of the Xk parameter is not clear.

Line 142:The format of the cited references does not appear to be the standard format in MDPFI

Figure 1,2,3: The color of scale numbering is dark grey, not black.

Figure 4: What is the meaning of “Legenda”? 

The result section needs more descriptions.

The conclusion is not concise and precise.

Reviewer 3 Report

1. Please check the scientific name of Pacific Oysters, the new genus name should be changed.

2. This work is meaningful and useful, if possible, we are interested in good and bad regions for oyster farming. Especially for the good areas, maybe it is better to highlight these areas with detailed figures (map). To across the seasons and fine-scale information.

3. In figure 6, it means oyster farming in a bad region require 812 days? but in the good region, it only needs 138 days. It is really like this? In my experiences, even in "poor" regions, it won't like this. Do you need to modify the parameter? Do any empirical data support these results? 

Round 2

Reviewer 2 Report

The paper has improved, but there are still some issues, such as:

  1. According to the results presented in the article, the title “Scaling up a model of pacific oysters (Crassostrea Gigas) growth in the Adriatic sea: implications for aquaculture spatial planning” may not be appropriate. Not to scale up the DEB model of oyster growth, but to apply the model in a large-scale area (Western Adriatic coastal areas), right?
  2. Please check the formula (1), it seems that a parenthesis is missing.
  3. Figure 1: The color of the label “2016”,”2017”,”2018”,”2019” are black, but the color of the text on the axis is dark gray. It is recommended to uniform their color, and the same for Figure 2 and Figure 3.

Author Response

  • According to the results presented in the article, the title “Scaling up a model of pacific oysters (Crassostrea Gigas) growth in the Adriatic sea: implications for aquaculture spatial planning” may not be appropriate. Not to scale up the DEB model of oyster growth, but to apply the model in a large-scale area (Western Adriatic coastal areas), right?

Thank you for the suggestion. the title was now changed to 'Testing a model of....'

  • Please check the formula (1), it seems that a parenthesis is missing.

Equation (1) is now been rewritten

  • Figure 1: The color of the label “2016”,”2017”,”2018”,”2019” are black, but the color of the text on the axis is dark gray. It is recommended to uniform their color, and the same for Figure 2 and Figure 3.

The figures have now been replaced